# In Silico Studies on Zinc Oxide Based Nanostructured Oil Carriers with Seed Extracts of *Nigella sativa* and *Pimpinella anisum* as Potential Inhibitors of 3CL Protease of SARS-CoV-2

**DOI:** 10.3390/molecules27134301

**Published:** 2022-07-04

**Authors:** Awatif A. Hendi, Promy Virk, Manal A. Awad, Mai Elobeid, Khalid M. O. Ortashi, Meznah M. Alanazi, Fatemah H. Alkallas, Maha Mohammad Almoneef, Mohammed Aly Abdou

**Affiliations:** 1Department of Physics, College of Science, Princess Nourah Bint Abdulrahman University, Riyadh 11671, Saudi Arabia; aahindi@pnu.edu.sa (A.A.H.); mmalenazy@pnu.edu.sa (M.M.A.); fhalkallas@pnu.edu.sa (F.H.A.); mmalmoneef@pnu.edu.sa (M.M.A.); 2Department of Zoology, College of Sciences, King Saud University, Riyadh 11451, Saudi Arabia; melobeid@ksu.edu.sa; 3King Abdullah Institute for Nanotechnology, King Saud University, Riyadh 11451, Saudi Arabia; mawad@ksu.edu.sa; 4Department of Chemical Engineering, King Saud University, Riyadh 11421, Saudi Arabia; ortashi9@ksu.edu.sa; 5Department of Physics, College of Sciences, University of Bisha, Bisha 61922, Saudi Arabia; m_abdou_eg@yahoo.com

**Keywords:** coronavirus, *Nigella sativa*, *Pimpenella anisum*, molecular docking, molecular dynamic simulation, zinc oxide nanoparticles

## Abstract

Coming into the second year of the pandemic, the acute respiratory syndrome coronavirus 2 (SARS-CoV-2) and its variants continue to be a serious health hazard globally. A surge in the omicron wave, despite the discovery of the vaccines, has shifted the attention of research towards the discovery and use of bioactive compounds, being potential inhibitors of the viral structural proteins. The present study aimed at the green synthesis of zinc oxide (ZnO) nanoparticles with seed extracts of *Nigella sativa* and *Pimpinella anisum*—loaded nanostructured oil carriers (NLC)—using a mixture of olive and black seed essential oils. The synthesized ZnO NLC were extensively characterized. In addition, the constituent compounds in ZnO NLC were investigated as a potential inhibitor for the SARS-CoV-2 main protease (3CLpro or Mpro) where 27 bioactive constituents, along with ZnO in the nanostructure, were subjected to molecular docking studies. The resultant high-score compounds were further validated by molecular dynamics simulation. The study optimized the compounds dithymoquinone, δ-hederin, oleuropein, and zinc oxide with high docking energy scores (ranging from −7.9 to −9.9 kcal/mol). The RMSD and RMSF data that ensued also mirrored these results for the stability of proteins and ligands. RMSD and RMSF data showed no conformational change in the protein during the MD simulation. Histograms of every simulation trajectory explained the ligand properties and ligand–protein contacts. Nevertheless, further experimental investigations and validation of the selected candidates are imperative to take forward the applicability of the nanostructure as a potent inhibitor of COVID-19 (Coronavirus Disease 2019) for clinical trials.

## 1. Introduction

The ongoing global pandemic continues to engulf the human population and still remains a grave health concern worldwide. Globally, there have been 490,853,129 confirmed cases of COVID-19, including 6,155,344 deaths reported to WHO in March 2022 [1]. It has been approximately two years since the WHO [1,2] declared COVID-19 as a Public Health Emergency of International Concern, followed by it being defined as a global pandemic in March 2020 [1,3]. Since then, there have been massive mortalities worldwide as the only countermeasures were the use of a broad spectrum of anti-viral drugs and the development of the vaccines. The dismal global situation still persists as there is a sudden surge of cases in different countries. This is attributed to the ability of the virus to rapidly mutate, resulting in highly transmissible variants that can cause re-infections, and also raises a concern on the efficacy of the current vaccines in containing the disease [4]. The major structural proteins of SARS-CoV-2 are the spike, envelope, membrane, and nucleocapsid proteins. In addition, the viral genome codes for 16 non-structural proteins (NSPs). In coronaviruses, NSPs play a vital role in RNA synthesis and processing, contributing to its survival as well as virulence. NSP5 is a 3C-like protease, and it directly mediates the maturation of NSPs, which is an essential part of the viral life cycle. Thus, 3CLpro makes an attractive target for anti-coronavirus drug development, where there is a substantial need to develop additional antiviral compounds with minimal side effects and alternate viral targets (other than ACE2-RBD interactions). Inhibitors targeting SARS-CoV-2 3CLpro mainly include peptide inhibitors and small-molecule inhibitors. Several natural compounds and derivatives with anti-virus and anti-inflammatory effects have exhibited a high binding affinity to 3CLpro. Thus, 3CLpro is an alternate target as it is a prominent protease and an essential component of the SARS-CoV-2 life cycle processing of the viral polyprotein [5,6,7]. The 3CLpro of COVID-19 exhibits 96% sequence similarity with that of SARS-CoV-2 [8]. Thus, 3CLpro inhibitors could be potential candidates as site specific anti-viral drugs. Considering the intricate pathophysiology of viral diseases coupled with the associated side effects of the current conventional drugs, scientists and pharmacologists worldwide have been urged to explore an alternative therapeutic approach, which is affordable in different sectors of the society. Plant extracts and phytochemicals have been exploited for their therapeutic benefits since time immemorial. This has led to several studies in the recent times that primarily targeted the evaluation of various phytochemicals to find potential inhibitors of Mpro through virtual screening and structure-based drug discovery approaches [9,10,11,12]. In the modern times of informatics, computational methods have been applied in the process of drug discovery, which has accelerated the discovery and design of novel drug candidates at a lower cost [13]. The critical role of in silico assays in the universal battle against COVID-19 has been enormous since the emergence of the disease. For COVID-19, these in silico tools have been beneficial for bioinformatics analysis of SARS-CoV-2 structures, analyzing potential drugs and introducing drug targets, also evaluating the efficacy of potential natural compounds for suppressing the COVID-19 infection. Additionally, aiding in the designing and optimizing of peptide-mimetic structures to extend them to clinical trials and repurposing the established and conventionally known therapeutics. These methods have facilitated the medical professionals and biotechnologists to design various vaccines, such as multiepitope vaccines using reverse vaccinology and immunoinformatics methods that have shown promising prospects through in vitro, in vivo, and clinical trial studies [14,15,16,17,18,19].

Natural products and essential oils are well recognized for their antiviral, anti-inflammatory, and compound-modulatory activities [20,21]. *Nigella sativa* is a flowering plant belonging to the Ranunculaceae family and has been extensively reported for its plethora of therapeutic benefits. *N. sativa* seeds contain unsaturated fatty acids (26–38%), proteins, alkaloids, saponins (melanin), and essential oils (0.4–2.5%). The major active constituents primarily include thymoquinone, dithymoquinone (nigellone), thymohydroquinone, thymol, and α-hederin. There have been recent reports that various phytochemicals obtained from plants, including the bioactive compounds in *N. sativa*, have the potential to inhibit the binding of viral spike (S) glycoprotein to the host cell [22]. Additionally, the hot water extract of seeds of *Pimpinella anisum* was reported to contain three lignin–carbohydrate–protein complexes with antiviral and immunostimulating activity. These complexes showed potent antiviral activities [23]. Further, the beneficial effects of *N. sativa* oil on immunity in microbial infection could be augmented by zinc (Zn) supplements. The possible therapeutic effects of *N. sativa* and Zn supplements against COVID-19 was suggested [24]. The potential efficacy of *N. sativa* oil (500 mg soft-gel capsules) complementing the standard treatment on the outcomes of patients with mild COVID-19 symptoms was also investigated [25].

Molecular docking and dynamic simulations are effective tools to assess dynamics of receptor–ligand binding affinity and interactions in drug discovery using nanomaterials. With this premise, the present study proposes a virtual screening-based drug discovery to assess the efficacy of pharmacologically active compounds in green synthesized ZnO NPs loaded nanostructured oil carriers (NLC) using a mixture of aqueous seed extracts of *Nigella sativa* and *Pimpinella anisum* with olive and black seed essential oils as potential inhibitors of Mpro.

## 2. Results

### 2.1. Characterization of Natural Lipid Carriers (NLC)

The UV-Vis absorption spectrum of the colloidal solution of ZnO NPs showed an absorption peak at around 369 nm (Figure 1, Appendix A). Figure 1A shows the UV-visible absorption spectrum of ZnO nanoparticles (ZnO NPs, red), while the essential oil capped ZnO NPs (ZnO NLC NPs, black) are shown in Figure 1A,B. The capping of ZnO NPs with essential oils shifted the UV visible absorbance of ZnO NLC NPs towards 369 nm (Figure 1B). The absorption peaks 369 and at 372 nm are the characteristic peaks for hexagonal wurtzite ZnO, respectively.

The photoluminescence properties and emission spectrum analysis of the synthesized ZnO NPs was carried out using spectrofluorophotometer. The emission spectrum exhibited two bands: one was in the UV region of 390–400 nm and the other was in the visible region of 420–650 nm The light emission and visible emission peaks were seen at 369 and around 450 nm, respectively (Figure 1C, Appendix A). The average particle size of synthesized ZnO NPs was found to be 149.7 nm and the polydispersity (PDI) index was 0.294 (Figure 1D, Appendix A). X-ray diffraction analysis was performed for phase verification and crystallinity of the produced zinc oxide nanoparticles. The diffractograms showed the peaks and XRD patterns are indexed as 32.15° (110), 34.39° (002), 36.20° (101), 37.84°, 47.49° (102), 56.52° (110), 62.79° (103), 67.88° (112), 69.01° (201), and 76.78° (202), respectively (Figure 2A, Appendix A).

The synthesized ZnO NPs was subjected to FTIR spectral analysis to evaluate the functional groups present in the synthesized nanoparticle and to confirm the presence of essential oil encapsulation on the nanoparticle. The FTIR spectrum of synthesized ZnO NPs and ZnO NP-loaded and -encapsulated nanostructured lipid carriers was recorded in the range of 4400–400 cm^−1^ (Figure 2B). It is observed that the bands were at 30,403.63 cm^−1^, 2353.96 cm^−1^ (C–H; bending vibration), 2180.91 cm^−1^ (–C=C–; bending vibration), 1444.36 cm^−1^ (C–C or C–O–H; bending vibration), 1117.89 cm^−1^, 1060.87 cm^−1^ (C–N; bending vibration), 866.96 cm^−1^ (N–H or C–H or C–Cl; bending vibration). Likewise, Figure 2B of ZnO NPs–NLC shows that the bands were at 3430.29 cm^−1^ (O–H or N–H; bending vibration), while the functional groups between 3000 cm^−1^ and 400 cm^−1^ suggest the possible groups from olive and black seed oils that were used in the encapsulated ZnO NPs; furthermore, broad absorption bands were observed around 600–400 cm^−1^, which could be attributed to ZnO stretching vibration. The TEM micrograph showed a less intense layering seen at the periphery of the nanoparticles (Figure 2C, Appendix A). The encapsulation was confirmed by the presence of spherical particles of capsular structure in the micrographs. Moreover, the micrographs indicate that the particles had a nanometer size with uniform distribution in the suspension without the formation of agglomerates. The micrographs showed a predominance of spherical nanoparticles with other varied shapes, such as rod-shaped and hexagonal (Figure 2C, Appendix A).

The energy dispersive X-ray diffractive (EDX) study was carried out for the synthesized zinc oxide nanoparticles to evaluate the elemental composition. The EDX spectrum shows strong major peaks between 0.5 and 9.6 keV (Figure 2D, Appendix A). EDX peaks showed that the synthesized sample of ZnO NPs are composed mainly of Zn, O, and Ca, attributed to compounds in the essential oils used, while the Cu element is associated with Cu–carbon support grids used in the analysis.

### 2.2. Molecular Docking

#### 2.2.1. Binding Energies

The docking scores of all bioactive compounds in the zinc oxide loaded nanostructure, natural lipid carriers (NLC), are presented in Table 1. The results indicate that the δ-hederin expressed the lowest binding energy (−9.9 kcal/mol) with the protein, followed by zinc oxide, oleuropein, and dithymoquinone. The findings suggest that among the constituents of the synthesized NLC, the phytochemicals in the *Nigella sativa* seed extract, δ-hederin and dithymoquinone, exhibited the lowest binding energy, along with zinc oxide and oleuropein in the olive oil. The above mentioned compounds with the best docking score were further selected for molecular dynamics simulations to validate the docking results.

#### 2.2.2. Quantitative Structure–Activity Relationship (QSAR) Studies

QSAR characterization and docking simulations are designed to investigate the potential of these selected compound protease inhibitors. Based on the QSAR properties, the four most stable compounds, owing to their low total energy, are thymoquinone, δ-hederin, oleuropein, and zinc oxide. These compounds also have a high surface area, solvent accessible surface area, volume, molar refractivity, and polarizability. They exhibited low dipole moment with good log P values, the highest being for dithymoquinone (Table 2).

#### 2.2.3. Molecular Dynamics Simulations (MDS)

##### Interaction Analysis of Protein

As shown in Figure 3, the 3D docking program detected three cavities in the protein to which the ligands did bind, where thymoquinone and oleuropein shared the same cavity (Figure 3). The results indicated that dithymoquinone expressed two hydrogen bonding interactions with residues Asn203 and His246. For oleuropein, four hydrogen bonds were observed with residues His246, Thr111, Gln110, and Asn151, while δ-hederin established two hydrogen-bonding interactions with residues Leu 287 and Lys 5. ZnO established four hydrogen interactions with residues Asn142, Met49, His163, and Gln189 (Figure 3, Figure 4 and Figure 5).

The details of the interactions are presented in Table 3. The prime MMGBSA method exhibited the relative binding-free energy (ΔG bind) of each optimized ligand molecule, and results are given in Table 4. The stability of the protein during simulation was assessed based on RMSD and RMSF parameters. Ligand properties were further explained by the four RMSD parameters: the radius of gyration (rGyr), molecular surface area (molSA), solvent-accessible surface area (SASA), and polar surface area (PSA) (Figure 6). The protein–ligand interactions, which were grouped into four categories, i.e., hydrogen bonding, hydrophobic, ionic, and water bridges, are illustrated by the protein–ligand and contact histograms (Figure 7).

The RMSD value for both the ligands and protein was calculated and noted from the graph and at the start of the simulation process for protein. However, massive fluctuations were observed up to 10 ns for dithymoquinone, which was followed by an equilibrium state 2.4 Å during simulation time. For the ligand, the RMSD value was noted to be 4.0–6.4 Å after half the time of the simulation, whereas the least fluctuations exhibited between 4.55 Å and 6.2 Å during the rest of the time (Figure 8A). For oleuropein, fluctuations were observed from 40–60 s. At the beginning of the simulation, the protein showed stable interaction between 1.2–2.4 Å. For the ligand, most of least fluctuations were observed from 4.0 Å to 4.8 Å (Figure 8B). δ-Hederin showed the most stable protein–ligand interaction with least and brief fluctuation after 80 ns. For the protein, least fluctuations were exhibited between 2.0 Å and 2.8 Å, while the ligand exhibited it between 9 Å and 15 Å (Figure 8C). Similarly zinc oxide showed only massive fluctuations after half time of the simulation (Figure 8D). Figure 8E represents the apo-form control for comparison.

The RMSF graph for the protein and ligand complexes and the apo-form control. is shown in Figure 9A–E. Each ligand exhibited exclusive fluctuations in various ranges. The overall RMSF values for all the four compounds was in the range of 0.5 Å to 2.5 Å. Least fluctuations for all the four compounds were in the range of 0.5 Å to 1.5 Å (Figure 9).

##### Ligand Properties

For dithymoquinone, the RMSD value stabilized in the range of 0.8 Å to 1.2 Å through the simulation time. The radius of gyration was in the range of 3.5 Å to 3.6 Å. The molecular surface area was noted in a range of 310 Å^2^ to 320 Å^2^. In addition, solvent accessible area exhibited fluctuations but was found to be constant in the range of 160 Å^2^ to 200 Å^2^. Similarly, the polar surface value showed fluctuations but reached an equilibrium state between 130 Å^2^ and 145 Å^2^ (Figure 6A). For oleuropein, the RMSD value stabilized in the range of 2.0 Å to 2.8 Å, while the radius of gyration was in the range of 6.0 Å to 6.4 Å. The molecular surface area was constant in a range of 490 Å^2^ to 5000 Å^2^. Furthermore, solvent accessible area exhibited initial fluctuations but was found to be constant in the range of 180 Å^2^ to 250 Å^2^. Similarly, the polar surface value showed fluctuations but reached equilibrium state between 260 Å^2^ and 275 Å^2^ (Figure 6B). For δ-hederin, RMSD value stabilized in the range of 1.0 Å to 1.6 Å, while the radius of gyration was in the range of 4.8 Å to 5.2 Å. Additionally, the molecular surface area exhibited a constant range between 490 Å^2^ and 496 Å^2^. Fluctuations were observed in the solvent accessible area but were constant over a range of 480 Å^2^ to 550 Å^2^, while the polar surface value reached the equilibrium state between 200 Å^2^ and 220 Å^2^ (Figure 6C). Lastly, for zinc oxide, the RMSD was stable throughout the simulation within a range of 0.2 Å to 0.25 Å with the radius of gyration stabilized between 4.20 Å and 4.24 Å. In addition, the molecular surface area, solvent accessible area, and polar surface values exhibited equilibrium states in the range of 324–332 Å^2^, 300–350 Å^2^, and 416–424 Å^2^, respectively (Figure 6D).

##### Protein Ligand Contacts

The protein–ligand contacts involved hydrogen bonds, hydrophobic bonds, and water bridges for most of the compounds evaluated. Among the four compounds, the ionic was observed only with the zinc oxide. For dithymoquinone, the protein–ligand involved hydrogen bonding with seven residues: Gln107, Pro108, Gln110, Asn203, Glu240, Asp245, and His246. For oleuropein, the H-bonds were formed with Gln107, Pro108, Gln110, Asn151, Val202, Glu240, and His246. For δ-hederin, Lys236, Tyr237, Asn238, Tyr239, Leu272, Met276, Asn277, and Ala285. For zinc oxide it was Glu166 (Figure 7).

The ligand–receptor interaction (histogram) includes two panels (Figure 10). The top panel illustrates all specific contacts of the protein with the ligand for each trajectory frame. From zero to nine, the contact numbers vary throughout the trajectory (Figure 10). Whereas the bottom panel analyzes the specific amino acid contributing towards the interaction with ligand (Figure 10). It is used to identify the specific amino acid involved in the interaction with ligand in each trajectory frame. The darker orange shade represents some amino acid residues involved in more than one precise contact with the ligand in a specific trajectory framework. The results are analogous to the histogram data represented in Figure 7. The protein–ligand interaction was stable and did not alter the structural conformation of the protein as shown by the Ramachandran plot (Figure 11).

## 3. Discussion

Since the unprecedented outbreak of the pandemic, there has been no standard therapy for the treatment of the disease. This has necessitated researchers and healthcare professionals worldwide to explore various drugs or alternative phytochemicals that could prove to be potent potential inhibitors of the viral proteins. In the recent times, nanomedicine has played a vital role in therapeutics, owing to a targeted drug delivery with minimal side effects and keeping green synthesis as a cornerstone of the approach. Several therapeutic phytochemicals structured as nanotechnology-based products are currently marketed and are under clinical investigation. The present study involved two phases which included the synthesis of zinc oxide nanoparticles—loaded nanostructured oil carriers (NLC)—using mixture of aqueous seed extracts of *Nigella sativa* and *Pimpinella anisum*, olive and black seed essential oils. UV-visible spectroscopic analysis aided to evaluate the changes that occurred in the UV-visible absorption of ZnO NPs after capping with nanostructured oil carriers over their surface. UV-Vis absorption spectrum of the colloidal solution of ZnO NPs was assigned an intrinsic band-gap absorption of ZnO NPs attributed to the electron transitions from valance to the conduction band [26]. This strong absorption in the UV region demonstrated that the synthesized ZnO NPs have good photocatalytic properties as well as novel applicability in medicinal applications [27]. After the capping of ZnO NPs with essential oils, a shift in the UV-visible absorbance of ZnO NPs–NLC was observed, which is explained due to the interaction of oils with ZnO NPs surface. This caused the disruption of the crystal lattice of ZnO NPs due to the presence of oils atoms in the lattice [28]. According to Gupta et al. [29], the absorption peak regularly shifts to the lower wavelength or higher energy with the decreasing size of the NPs. According to Mei’s theory, the size of the nanoparticles is spherical if there is only one sharp absorbance peak in the UV spectrum. In the present study, the synthesized loaded NLC were mostly spherical in shape [30,31]. The photoluminescence analysis results of the synthesized ZnO NPs exhibited sharp UV emission peaks, which was attributed to the recombination of electrons in the conduction band and holes in the valance band [32]. Almost all ZnO morphologies possess two emission bands at room temperature: a near band-edge light emission and a broad, deep-level (visible) emission [33,34]. The strong band-edge peak resulted from excitonic recombination correlated with the near band edge emission, while the visible emission peak occurred owing to defects, such as oxygen vacancies that are responsible for broad-band emission [35].

The polydispersity index (PdI) indicates that synthesized nanoparticles are monodispersed, which ensured that the synthesized ZnO NPs have good stability [36]. X-ray diffraction results showed the patterns of reflection peaks of ZnO NPs (COD 2300113), confirming and corresponding to the characteristic hexagonal wurtzite structure of synthesized ZnO NPs. The presence of sharp and intense diffraction peaks indicates the high crystalline nature of ZnO NPs, while the other peaks could be from the essential oils that were used in capping the ZnO NPs.

The FTIR results of synthesized ZnO NPs indicated the formation of the metal–oxygen stretching of the zinc oxide nanostructure. Saraswathi et al. [37] showed that the region between 400 and 600 cm^−1^ is the Zn–O group. In addition, Murugan et al. [38] also observed a high density band around 440 cm^−1^ caused by the zinc and oxygen bond stretching mode. The functional groups between 3000 cm^−1^ and 400 cm^−1^ suggest the possible functional groups from olive and black seed oils involved in the encapsulation of the NPs; furthermore, broad absorption bands were observed around 600–400 cm^−1^, which could also be attributed to the Zn–O stretching vibration [39].

From the electron micrographs it was possible to observe an organic layer between the particles, which is attributed to the presence of essential oils on the surface of ZnO NPs. This layer is responsible for stabilization and prevention of agglomeration and precipitation of the NPs [40]. The EDX spectrum showed strong major peaks, which confirmed the elemental distribution of the ZnO NPs [41].

The study also included an investigation of the constituent phytochemicals and zinc oxide in the NLC as potential inhibitors of 3CL protease of SARS-CoV-2 through virtual screening and structure-based drug discovery approaches. The viral structural protein 3CLpro (PDB ID 6M2Q) was selected as the therapeutic target [42]. In all, the 27 bioactive compounds in the NLC, along with the zinc oxide, were used for molecular docking studies. All the compounds interacted with the binding pocket cavity of the protease. Although the phytochemicals in *Pimpinella* seed extract also showed moderately good binding energies (estrol; −7.6 kcal/mol, estrone; −7.4 kcal/mol) four compounds with the best docking score were selected for further simulation studies. Thymoquinone, hederin (δ-hederin) from the *Nigella* seed extract, oleuropein from olive oil, and zinc oxide showed the best docking scores, indicating a more stable interaction with the protease. The top scored compounds in this study based on the docking score against the 3CLpro were in the order of δ-hederin > zinc oxide > oleuropein > dithymoquinone. The range of the docking score was −9.9 kcal/mol to −7.9 kcal/mol, which was close to some bioactive compounds reported in a recent study [43]. Active compounds with ΔG less than −8.0 kcal/mol, namely quinine, quinidine, cinchonine, and lovastatin, were selected in a previous study with recombinant 3CL assays by Haniyya et al. [43]. The other constituent compounds from the *Nigella* and *Pimpenella* seed extracts exhibited moderate binding energies. Several studies on the extracts and essential oil of *Pimpinella anisum* have reported the presence of constituent chemical compounds with pharmacological properties, such as antimicrobial, antifungal, antiviral, antioxidant, and insecticidal effects [23]. A recent in silico study on the phytochemicals of *P. anisum* as potential inhibitors of SARS-CoV-2 3C-like protease showed that out of eleven phytochemicals evaluated, five constituent compounds illustrated a binding energy of 7.9 kcal/mol. This is in line with the results of the docking score of the present study and merits further studies both in vitro and in vivo to evaluate their ability to inhibit SARS-CoV-2 and their use in therapeutics. Islam et al. [44] reviewed the pharmacological properties of *N. sativa* seed and thymoquinone, including immunomodulatory, antioxidant, and anti-inflammatory effects and their potential therapeutic strategy against COVID-19. Extensive studies have reported that thymoquinone has a high to moderate binding affinity to the recognition site for the SARS-CoV-2 spike. Thus, this bioactive compound could be a candidate compound to reduce the risk of COVID-19 or possibly to treat the disease, especially in people with comorbidities [45]. Additionally, it has been reported that the binding affinity of a *N. sativa* constituent, dithymoquinone, was higher than a positive control (chloroquine), which has a high potential affinity binding at the SARS-CoV-2–ACE2 interface. These reports are in consensus with the findings of the present study as among the bioactive constituents in the zinc oxide NLC, thymoquinone showed a high binding potential at the SARS-CoV-2–3CLpro interface. Furthermore, another constituent of *N. sativa* in the ZnO NLC δ-hederin; a derivative of α-hederin also showed a high affinity binding potential. Previously, constituents of *N. sativa*, such as α-hederin, thymohydroquinone, and thymoquinone, have shown efficient binding to ACE2, which suggests a potential therapeutic effect of these bioactive components to combat COVID-19 [46,47].

The four optimized compounds of interest were then used for the QSAR study, a methodology that allows an appropriate validation of biological activities from the molecular structure of chemical compounds. The QSAR data clearly showed that coefficient (log P) value for dithymoquinone was highest, followed by oleuropein and δ-hederin. Log P values are important to assess the biological activity of compounds and estimate their cell membrane permeability [4,48]. The optimized compounds had low dipole moments and good log P values as well, which reflected good binding interactions [49].

Docking studies primarily determine the binding pose of the ligand to the protein, while MD simulation focuses more on molecular flexibility, which complements the docking study. The root mean square deviation (RMSD) and root mean square fluctuation analysis along the trajectory of simulation demonstrates a quantitative measure of the stability of the protein–ligand complex. A wide persistent fluctuation in RMSD and RMSF trajectory reflects conformational changes, including readjustments in the protein–ligand system [50]. All the optimized compounds used in the present study did show minor ranges of initial/later fluctuations but were equilibrated over a long range, indicating stability. Taken together, the four compound–protein complexes seem to be fairly stable as the RMSD values were observed to be far less than 3.0 Å, which is considered stable [51]. The high RMSF value shows more flexibility, whereas low RMSF value shows limited movements. In the present study, the RMSF was less than 3.0 Å for all the optimized compounds throughout the simulation. The values are close to ideal value of 3.4 Å [52].

The stability of the protein–ligand complex is primarily due to the non-covalent interactions (hydrophobic contacts, hydrogen bonds, water bridges, salt bridges) between proteins and ligands, which in turn significantly impact the resulting binding energy [51]. In the present study, the MD simulation showed that dithymoquinone, δ-hederin, and oleuropein exhibited maximum water bridges, along with hydrogen bonding and hydrophobic interactions, which reflected a stable ligand–protein interaction. These binding cavities are located near the substrate-binding active site. This is best explained by the substrate-binding pocket represented by an intermolecular surface in a study by Su et al. [53]. It was reported that the protease had a catalytic Cys145–His41 dyad and an extended binding site, features shared by both SARS-CoV-2 3CLpro and MERS-CoV 3CLpro. The examination of the active site of the complex revealed that three phenolic hydroxyl groups made multiple hydrogen bonds with the main chains of Leu141/Gly143 as well as the side chains of Ser144/His163. The only carbonyl group established a hydrogen bond with Glu166, and hydrophobic interactions were found with multiple residues Gln189/Arg188/Met49/Cys44/His41. Similar, ligand–protein interactions with the abovementioned amino acids were observed in the present study as well. Thus, the inhibition is not allosteric but competitive.

Furthermore, the Ramachandran plot analysis showed that no secondary structural changes occurred in the protein during binding to the four optimized ligands. Among the active bioactive constituents from *N. sativa*, thymoquinone has been given more emphasis as a potential inhibitor of ACE2 of SARS-CoV-2 in most of the studies. The α-hederin and its derivatives, such as δ-hederin found in *N. sativa*, were paired with classical biological activities, such as antioxidant, anti-inflammatory, anti-microbial, anti-bacterial, anti-fungal, anti-viral [54]. Additionally, in recent times, Zn has been widely reported to boost the immune system and balance the immune response during viral infections, including SARS-CoV-2. The immune-boosting activities of Zn have been due to the proliferation and activation of neutrophils, NK cells, macrophages, and T and B cells as well as cytokine production. Zinc also ameliorates the adverse effect of ROS that contribute to the inflammatory processes [24].

Zinc oxide nanoparticles have been reported as antiviral agents. Furthermore, a non-toxic environmentally benign green platform for drug delivery would be ideal and safe. Thus, the use of phytochemicals derived from natural sources have shown that bioactive compounds exhibit strong virucidal effect against coronaviruses. The nanoflora has been proven to enhance the bioavailability of these bioactive compounds to facilitate effective delivery of insoluble phytochemicals, repurposing antimicrobial drugs and thus their therapeutic approach can be utilized to combat against viruses and coronaviruses (CoVs) [55]. In congruence with this, the in silico studies showed a marked inhibitory potential of thymoquinone, δ-hederin, oleuropein, and zinc oxide against the 3CL protease of SARS-CoV-2. Thus, the key findings of the present study clearly describe the potential of the nanostructure ZnO NLC, synthesized using the seed extracts of *N. sativa* and *P. anisum* with an encapsulation of the essential oils of olive and black seeds as a stable therapeutic agent against the 3CL protease of SARS-CoV-2.

## 4. Materials and Methods

### 4.1. Preparation of the Seed Extract

The seeds of *Nigella sativa* and *Pimpinella anisum* were purchased from an herbal medicine store in Riyadh, Saudi Arabia. The seeds were identified by the Department of Botany of Science at the Faculty of Science, King Saud University, KSA. With an electrical grinder, *N. sativa* and *P. anisum* seeds were grounded into powdered materials and then powdered. To obtain the aqueous extract of *N. sativa* and *P. anisum*, 100 g of the obtained powdered materials were dissolved in 400 mL of distilled water and then were kept in a refrigerator for 24 h. The extract was then filtered and kept for use in further studies.

### 4.2. Synthesis of Zinc Oxide Nanoparticles

Zinc oxide nanoparticles (ZnO NPs) were synthesized by the precipitation method. Two grams of zinc nitrate hexahydrate (Zn(NO_3_)_2_·6H_2_O, Loba Chemie, Mumbai, India) was added to the 50 mL of prepared seed extract under continuous heating and stirring at (500 rpm/70 °C) for 2 h to assist the electrostatic interaction of Zn^2+^ with the extract biomolecules. Thereafter, freshly prepared (2 M) sodium hydroxide (NaOH, Qualikems Fine Chem Pvt. Ltd., Mumbai, India) solution was added drop-wise under magnetic stirring. The reactants were continuously stirred for 2 h, to obtain greyish suspension. The formation of ZnO NPs was indicated by the appearance of the greyish precipitate, which was centrifuged at 10,000/10 (rpm/min) and subsequently washed with ethanol and deionized water to remove the impurities from the surface of the synthesized ZnO NP. This was followed by drying at 90 °C in an oven for 24 h. The calcination of the dried ZnO NPs was carried out at 400 °C for 4 h in a furnace (Carbolite, CWF1300, Hope, UK), followed by making a fine light grey color powder of ZnO NPs, this fine powder of green synthesized ZnO NPs was used for further characterization. Synthesized zinc oxide nanoparticles—loaded nanostructured oil carriers (NLC)—were synthesized using mixture of olive and black seed essential oils according to a modified method given by Shajari et al. [56] and Ali et al. [57].

### 4.3. Characterization of Nanoparticles

The absorption spectra of the colloidal solution of synthesized ZnO NPs were obtained in the wavelength range 100–800 nm using an ultraviolet-visible (UV-Vis) UV 2450 spectrophotometer. The photoluminescence properties of ZnO NPs were measured by spectrofluorometer (Shimadzu, RF-6000, Equipnet, Tokyo, Japan). The dynamic light scattering (DLS) technique based on the function of time was carried out using a Zetasizer (HT Laser, ZEN3600 Malvern Instruments, Malvern, UK). The presence of surface functional groups and the binding resistance of the synthesized ZnO NPs was analyzed and established by using Fourier transform infrared spectroscopy (FTIR) using a BX spectrometer (Perkin Elmer, Waltham, CA, USA) in the range of 4000–400 cm^−1^. The crystalline nature and purity of the synthesized ZnO NPs was determined by using Bruker D8 ADVANCE X-ray diffractometer (Bruker, Billerica, MA, USA) operating at 40 kV and 40 MA with CuKa radiation at 1.5418 Å in the 2θ range of 0–80°. The surface morphology and structure of the ZnO NPs was confirmed by TEM analysis performed on a TEM, JEM-2100F (JEOL Ltd., Peabody, MA, USA) operated at an accelerating voltage of 200 kV. The Brunau–Emmet–Teller (BET) method was employed to measure the surface area of synthesized ZnO NPs with a Gemini 2360 surface area analyzer (Micrometrics, Norcross, GA, USA).

### 4.4. Protein Targets and Ligands

Receptors: Crystal structure of 3CL protease (PDB ID: 6M2Q).

Ligands: Bioactive compounds in zinc oxide loaded nanostructure; natural lipid carriers (NLC) using a mixture of *Pimpinella anisum* and *Nigella sativa* seed extracts (Table 5, Appendix A).

The standalone, offline software used was the Molecular Operating Environment, 2015 version (Chemical Computing Group, Montreal, QC, Canada) and the Discovery Studio 2019 Client Full Package (Biovia, San Diego, CA, USA). The molecular docking study and the molecular dynamic simulation was performed in accordance with the methods used in a previous study by Ononamadu et al. [51].

### 4.5. Molecular Docking Study

Molecular docking was used to predict the binding interactions between each protein and the selected phytochemicals (ligands). The three-dimensional crystal structure of SARS-CoV-2 Mpro (PDB ID: 6M2Q) were used as the biological targets for the docking analysis. We investigated twenty-eight constituent phytochemicals in the ZnO NLC. All phytochemicals interacted with the main protease (Mpro). The best docking (binding free energy) scores for all phytochemicals were found in the range −7.9 to −9.9 kcal/mol (Table 1).

#### QSAR Studies

QSAR studies were used to anticipate the reactivity and properties of the selected compounds. The computational calculation was carried out using the HyperChem Professional 8.0.3 program (Hypercube, Gainesville, FL, USA). Initially, the compounds with a good docking score were optimized using the (MM+) force field, with semi-empirical PM3 methods, and energy minimization was achieved using a Fletcher−Reeves conjugate gradient algorithm method.

### 4.6. Molecular Dynamics Simulation

The four optimized compounds were chosen as the best ligands for further MD simulation work. A molecular dynamics simulation of 100 ns was carried out for these compounds to obtain better insight into the stability of the protein–ligand complexes. A ligand docking module of Maestro 12.3 was used to carry out molecular docking simulation studies. Before docking, the ‘Receptor Grid Generation’ module was used to generate the active binding site on the target protein. While doing so, the van der Waals radius scaling factor and partial charge cut-off was kept at 1.0 and 0.25, respectively. The rest of the remaining parameters were kept as default. The ‘Extra Precision’ (XP) model was used to perform the molecular docking of the 27 phytocompounds to the active binding site of SARS-CoV-2 3CL protease (PDB ID: 6M2Q). The binding affinity and molecular interaction behavior were obtained. MD simulation studies were carried out with Schrodinger’s Desmond module. Before the MD simulation was started, an equilibration process was carried out till the system reached a stationary state. The MD simulation was carried out at a temperature and an ambient pressure of 310 K and 1.013 bar, respectively, for a period of 100 ns. The results of the MD simulation were extensively analyzed with a simulation interaction diagram. The root mean square deviation (RMSD) of the protein−phytocompound complex, the root mean square fluctuation (RMSF) of the protein, the protein−ligand interaction diagram, the interacting amino acid residues with the ligand in each trajectory frame, and the trajectory of different ligand properties were analyzed. The overall stability of the protein was further investigated through Ramachandran plot analysis.

## 5. Conclusions

The study presented the stability and flexibility changes in SARS-CoV-2 WT 3CL protease by using MD simulation. The interaction patterns with thymoquinone, δ-hederin, oleuropein, and zinc oxide showed some variation. Thus, all the compounds with the best docking score had some flexibility that can affect function and catalytic activities. Understanding the flexibility of the 3CL protease protein of SARS-CoV-2 and its interaction with known and approved drugs can help to improve the design and development of new drugs. This merits the potential use of ZnO NLC as a therapeutic strategy on its own as well as to complement the conventional counter treatments for COVID-19. However, based on the prospective therapeutic benefits of the ZnO NLC, future in vitro and in vivo studies are imperative to assess the efficacy of the nanostructure as a whole in biological systems. Extensive further studies would help in validating the results of the computational analysis and take it forward. Furthermore, the key findings of the study could possibly address a nanotechnological approach with anti-coronavirus abilities to provide researchers with insightful strategies directed towards developing novel antiviral therapies to prevent pandemics similar to the current COVID-19 in the near future.

## Figures and Tables

**Figure 1 molecules-27-04301-f001:**
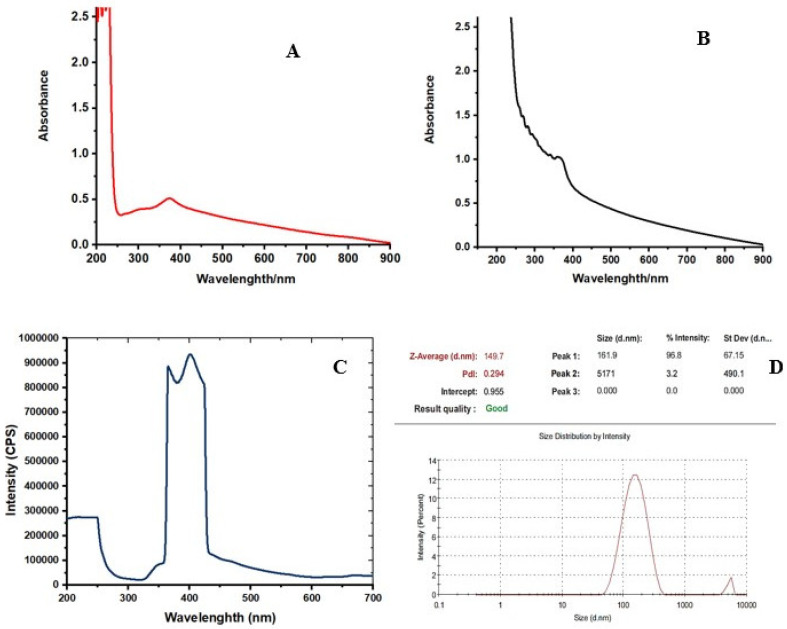
The UV-visible absorption spectra of synthesized (**A**) ZnO NPs (**B**) ZnO NPs–NLC. (**C**) the emission spectrum of synthesized ZnO NPs. (**D**) Particle size and polydispersity index of the ZnNPs.

**Figure 2 molecules-27-04301-f002:**
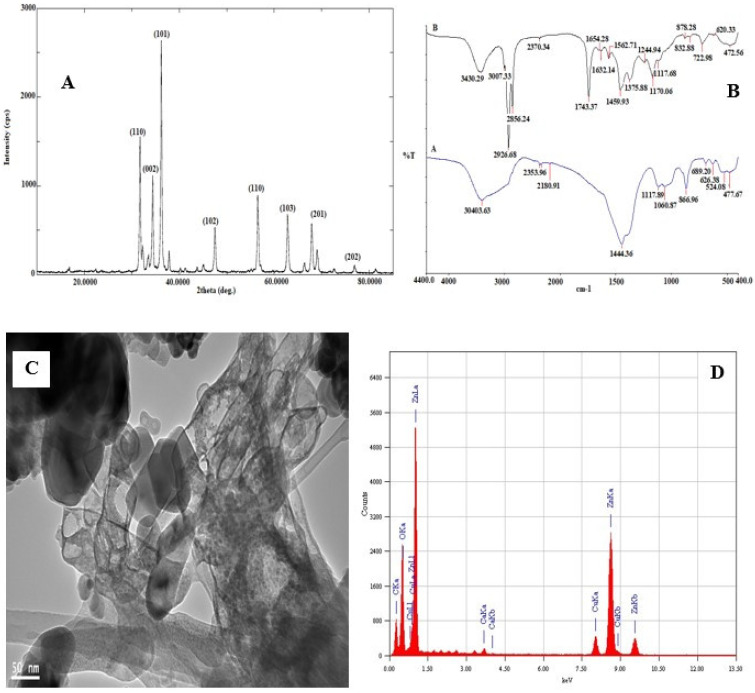
(**A**) X-ray diffraction pattern and diffraction angles peaks of synthesized ZnO NPs–NLC. (**B**) FT-IR spectra of synthesized (**A**) ZnO NPs (**B**) ZnO NPs–NLC. (**C**) TEM micrograph and (**D**) EDX spectrum of synthesized ZnO NPs.

**Figure 3 molecules-27-04301-f003:**
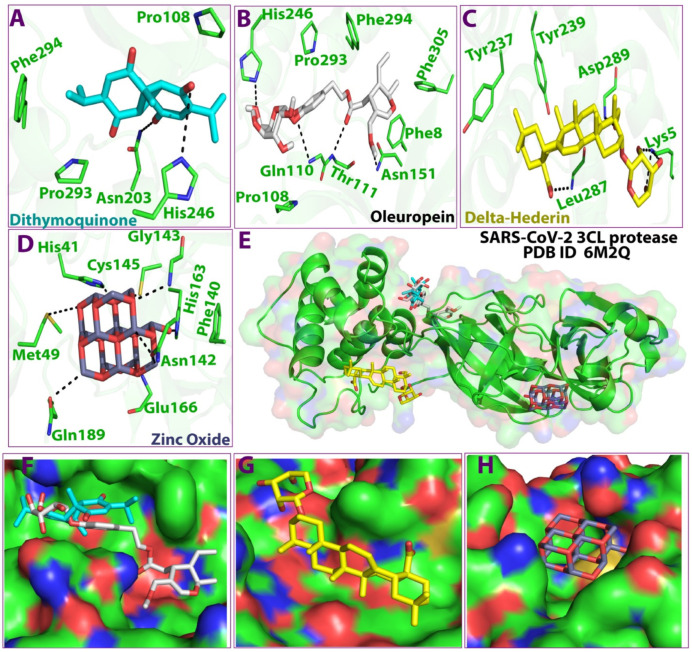
3D Ligand–receptor interaction map of top scored compounds in the zinc NLC with SARS-CoV-2 (3CLpro). (**A**) dithymoquinone, (**B**) oleuropein, (**C**) delta-hederin (δ-hederin), (**D**) zinc oxide, (**E**) interaction map showing all four compounds, (**F**) binding cavity of the protein showing dithymoquinone and oleuropein, (**G**) binding cavity of the protein showing delta-hederin, (**H**) binding cavity of the protein showing zinc oxide.

**Figure 4 molecules-27-04301-f004:**
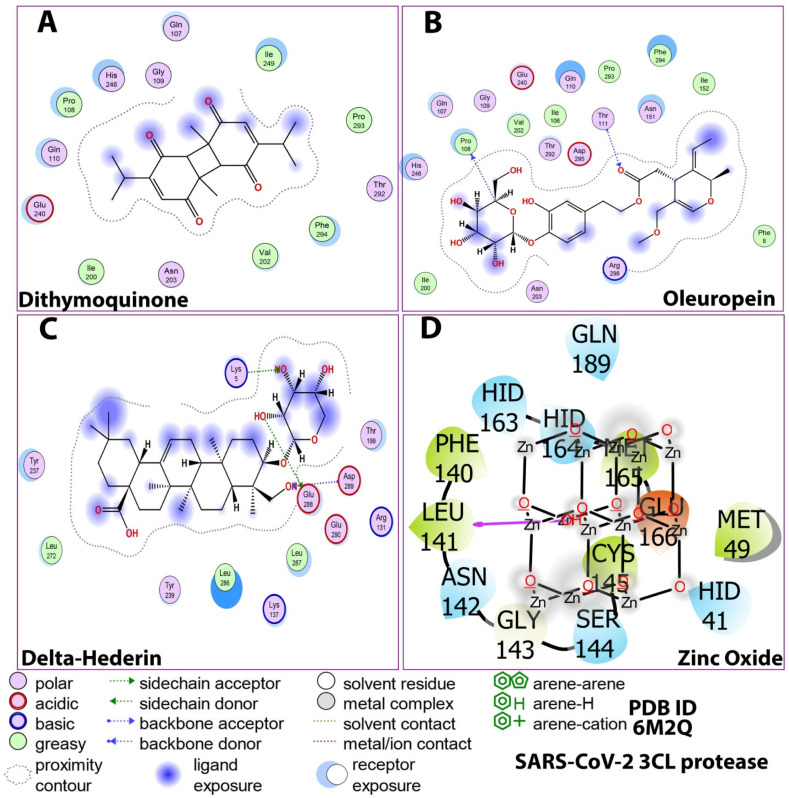
Post-docking compound–protein interaction map displayed in 2D showing the nature of molecular interaction between the ligands and the protein. (**A**) dithymoquinone, (**B**) oleuropein, (**C**) delta-hederin (δ-hederin), (**D**) zinc oxide.

**Figure 5 molecules-27-04301-f005:**
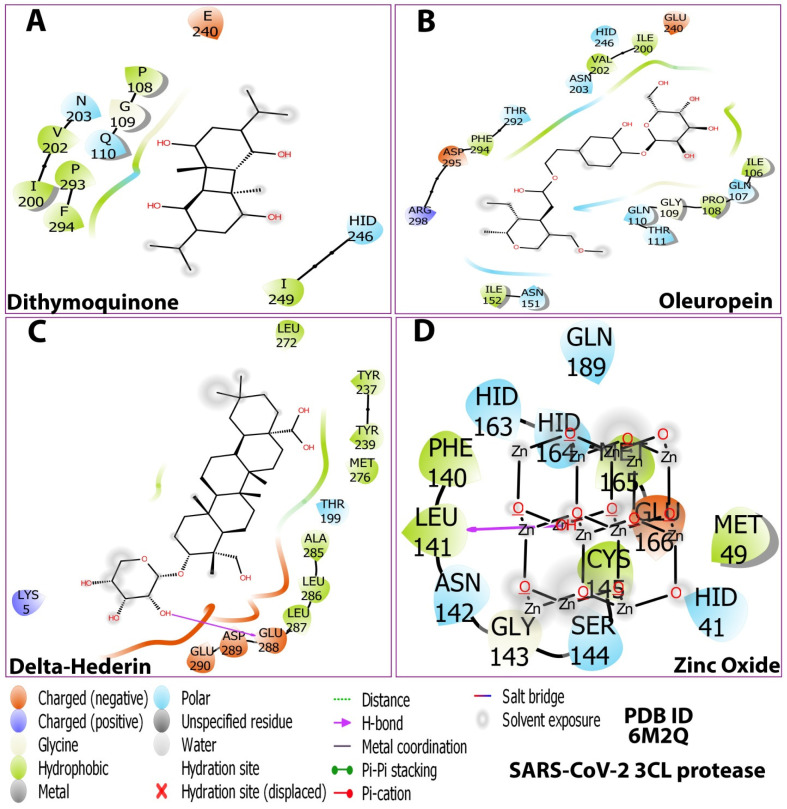
2D Ligand–receptor interactions of optimized compounds in the zinc NLC with SARS-CoV-2 (3CLpro) displaying molecular interactions between the ligands and the amino acids of the protein. (**A**) dithymoquinone, (**B**) oleuropein, (**C**) delta-hederin (δ-hederin), (**D**) zinc oxide.

**Figure 6 molecules-27-04301-f006:**
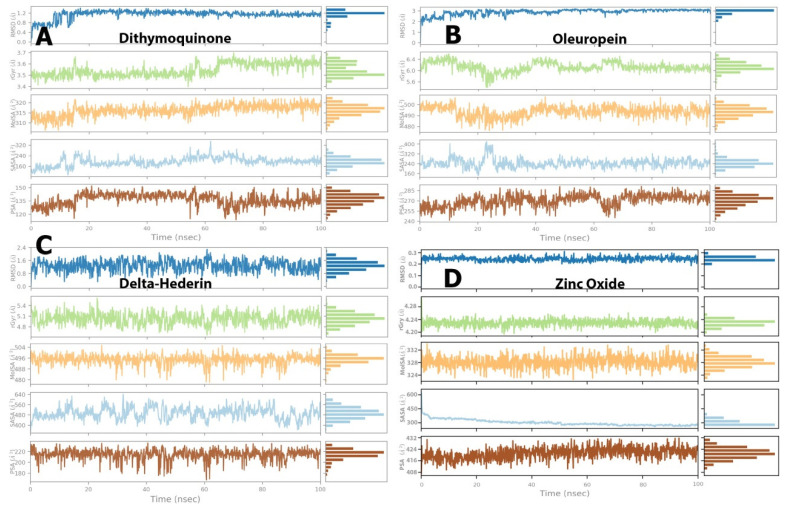
Post-molecular dynamics simulation analysis of protein and ligand properties, radius of gyration (rGyr), molecular surface area (MolSA), solvent accessible surface area (SASA) and polar surface area (PSA). (**A**) dithymoquinone, (**B**) oleuropein, (**C**) delta-hederin, (**D**) zinc oxide.

**Figure 7 molecules-27-04301-f007:**
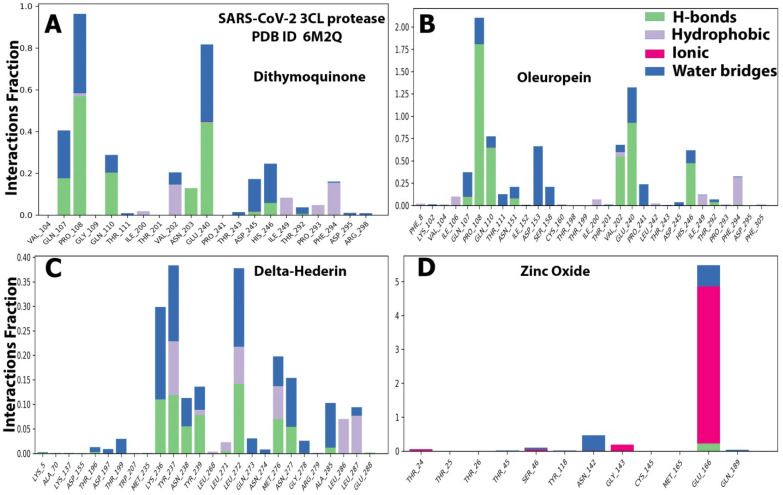
The histograms of protein ligand contact based on the nature of molecular interactions of the four optimized compounds. (**A**) dithymoquinone, (**B**) oleuropein, (**C**) delta-hederin, and (**D**) zinc oxide throughout the trajectory.

**Figure 8 molecules-27-04301-f008:**
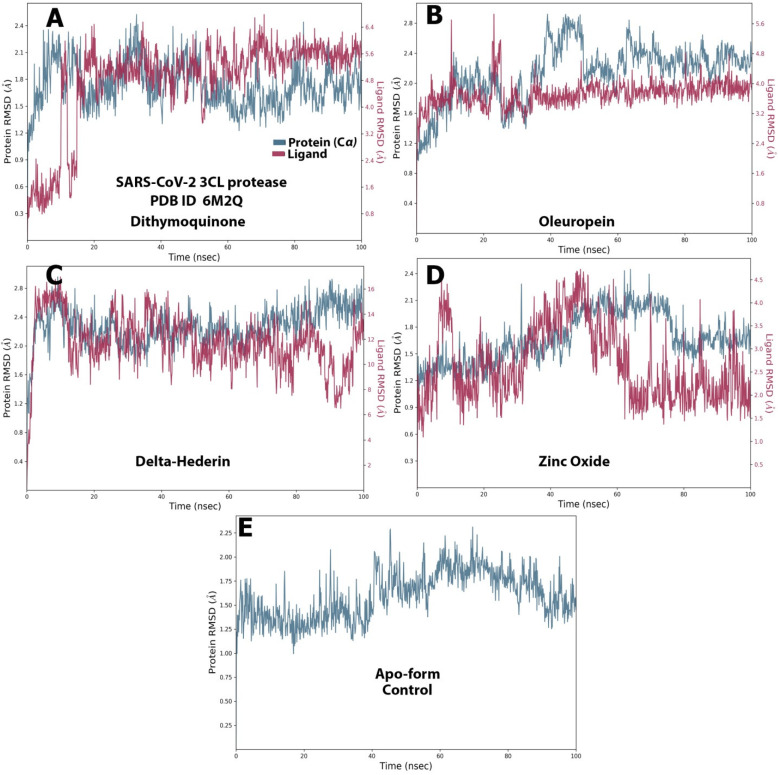
The protein–ligand RMSD of the optimized compounds during the 100 ns. (**A**) dithymoquinone, (**B**) oleuropein, (**C**) delta-hederin, (**D**) zinc oxide, (**E**) apo-form as a control.

**Figure 9 molecules-27-04301-f009:**
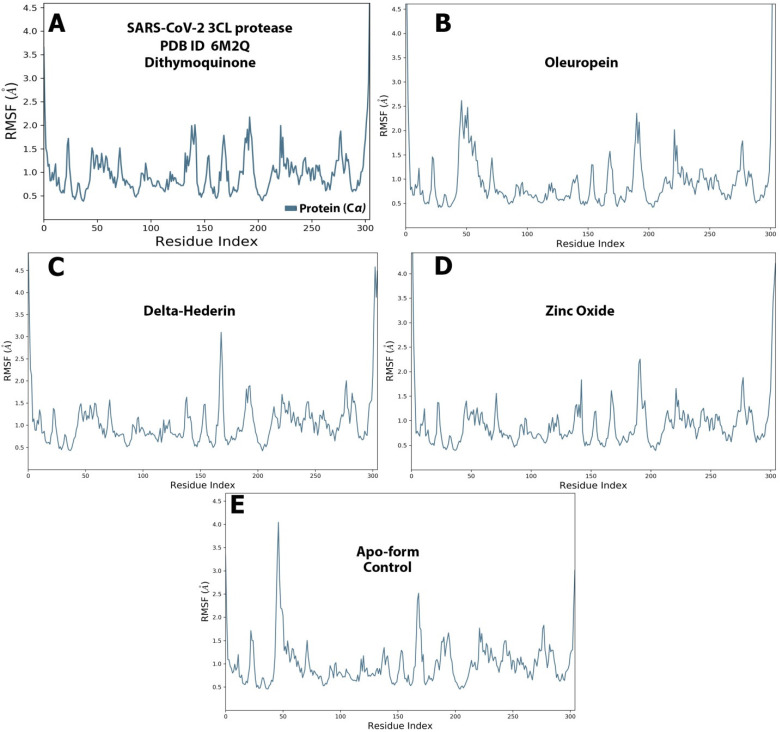
The protein–ligand RMSF of the optimized compounds during the 100 ns. (**A**) dithymoquinone, (**B**) oleuropein, (**C**) delta-hederin, (**D**) zinc oxide, (**E**) apo-form as a control.

**Figure 10 molecules-27-04301-f010:**
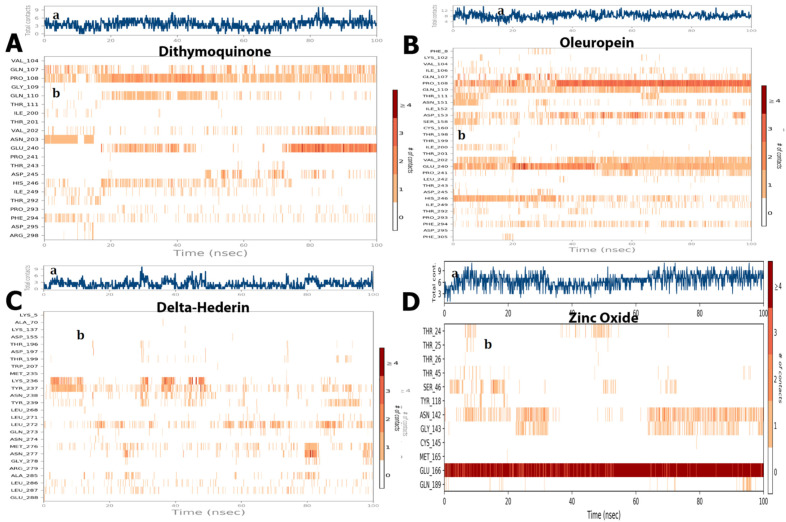
(a) The total number of contacts/interactions in each trajectory framework of four optimized compounds. (**A**) dithymoquinone, (**B**) oleuropein, (**C**) delta-hederin, (**D**) zinc oxide. Interaction is represented by the effective site of amino acids in each trajectory framework of the four optimized four compounds. (b) Hydrogen bonding interaction (% age) of residue Glu166 during the simulation.

**Figure 11 molecules-27-04301-f011:**
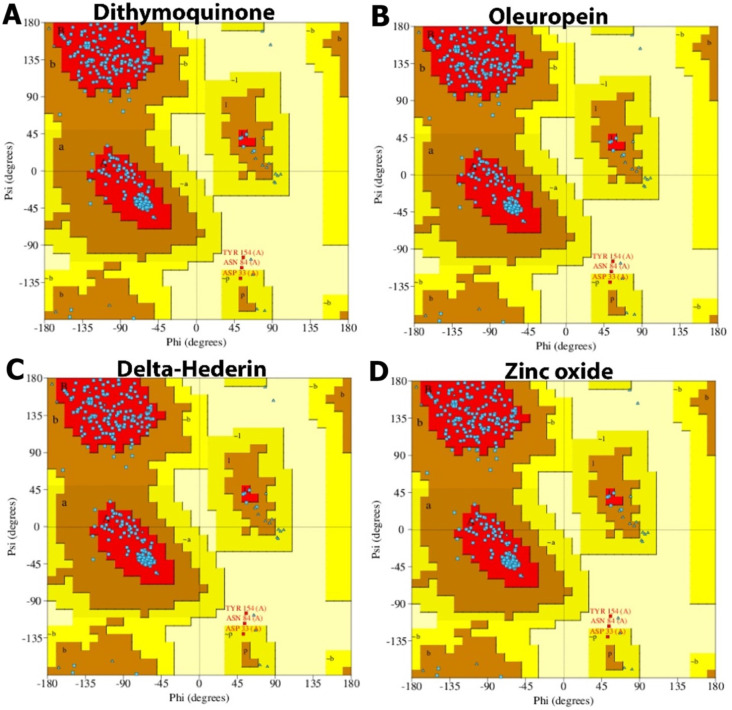
Ramachandran plot of protein interacting with the best docked compounds. (**A**) dithymoquinone, (**B**) oleuropein, (**C**) δ-hederin, (**D**) zinc oxide.

**Table 1 molecules-27-04301-t001:** Molecular docking results; binding energies of constituent compounds.

Compound	Binding Energy (kcal/mol)
4-terpineol	−5.4
Anisaldehyde	−4.6
Carvacrol	−5.6
Coumarin	−5.5
Dithymoquinone	−7.9
Estragole	−5.0
Estrol	−7.6
Estrone	−7.4
Eugenol	−5.3
δ-Hederin	−9.9
Hydroxytyrosol	−5.4
Methylchavicol	−6.0
Nigellone	−5.4
Oleocanthal	−6.2
Oleuropein	−8.4
*p*-Cymene	−5.3
Scopoletin	−5.8
Sesquiterpene-Himachalene	−6.7
Sesquiterpene-longifolene	−6.6
*t*-Anethole	−5.3
Thymohydroquinone	−5.5
Thymol	−5.3
Thymoquinone	−5.3
*t*-Anethol	−5.3
Tyrosol	−5.1
Umbelliferone	−5.8
Zinc Oxide	−8.8

**Table 2 molecules-27-04301-t002:** QSAR data for optimized compounds.

Function	Dithymoquinone	Oleuropein	δ-Hederin	Zinc Oxide
**Surface area (compound) (Å^2^)**	287.26	761.95	680.82	605.19
**Surface area (Grid) (Å^2^)**	510.16	772.18	757.99	465.81
**Volume (Å^3^)**	857.42	1317.98	1399.58	752.79
**Hydration energy (kcal/mol)**	−5.00	−35.71	−25.96	−101.84
**Log P**	6.35	4.15	3.57	0.50
**Refractivity (Å^3^)**	21.58	62.94	87.31	5.52
**Polarizability (Å^3^)**	20.48	39.28	43.53	2.97
**Mass (amu)**	304.22	493.32	553.42	1058.81
**Total energy (kcal/mol)**	55.7699	11.6418	35.8383	187.262
**Dipole Moment (Debye)**	0	1.8	1.15	1.115
**RMS Gradient (kcal/Å mol)**	0.08899	0.116	0.1357	0.08826

**Table 3 molecules-27-04301-t003:** Interactions and binding energies of ligands with SARS-CoV-2 3CL protease (PDB ID: 6M2Q).

Sl. No.	Ligand	Receptor-Chain (A)	Interaction	Distance	E (kcal/mol)
**Oleuropein**	C22 32	O PRO108	(A) H-donor	3.42	−0.7
	O2 11	N THR 111	(A) H-acceptor	3.26	−3.3
	O2 11	OG1 THR111	(A) H-acceptor	3.24	−0.8
**δ-Hederin**	O6 36	OE1 GLU288	H-donor	3.02	−2.7
	O8 38	NZ LYS 5	(A) H-acceptor	2.92	−6.3
	O2 43	N ASP 289	(A) H-acceptor	3.39	−1.1
**Dithymoquinone**			hydrophobic interaction	4	0.0
**Zinc Oxide**			Ionic and hydrophobic interaction	4	0.0

**Table 4 molecules-27-04301-t004:** Binding energies (MMGBSA) of the complexes of matrix protein of 3CL protease and the selected compound.

Compound	MMGBSA dG Bind (kcal/mol)	MMGBSA dG Bind Coulomb (kcal/mol)	MMGBSA dG Bind Covalent (kcal/mol)	MMGBSA dG Bind Hbond (kcal/mol)	MMGBSA dG Bind Lipo (kcal/mol)	MMGBSA dG Bind Solv GB (kcal/mol)	MMGBSA dG Bind vdW (kcal/mol)
Oleuropein	84.54333529	33.26906716	1.250663764	4.130708122	24.65461085	28.35451159	52.09412451
δ-Hederin	0.09837639	−0.254694843	0.09837639	−10.3479	−10.34134391	10.74413339	−20.18357659
Zinc Oxide	688.1897843	−1668.717611	−52.87536451	−0.013621852	−0.013621852	2413.661494	−3.86511158
Dithymoquinone	−26.19458915	5.244208157	−1.027737214	−0.026064154	−12.49690125	13.12157949	−31.00967417

**Table 5 molecules-27-04301-t005:** Constituent compounds in zinc oxide loaded nanostructure; natural lipid carriers (NLC).

No.	Constituents	Compounds
1	*Pimpinella anisum*	eugenol, *t*-anethole, methylchavicol, anisaldehyde, estragole, coumarins, scopoletin, umbelliferon, estrols, terpene hydrocarbons, polyenes, and polyacetylenes
2	*Nigella sativa*	thymoquinone (30–48%), thymohydroquinone, dithymoquinone, *p*-cymene (7–15%), δ-hederin, carvacrol (6–12%), 4-terpineol (2–7%), *t*-anethol (1–4%), sesquiterpene longifolene (1–8%) α-pinene and thymol
3	Zinc oxide nanoparticles	
4	Olive oil	Tyrosol, hydroxytyrosol, oleocanthal, oleuropein

## Data Availability

Not applicable.

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
