# Peer review of "In Silico Studies on Zinc Oxide Based Nanostructured Oil Carriers with Seed Extracts of Nigella sativa and Pimpinella anisum as Potential Inhibitors of 3CL Protease of SARS-CoV-2"

_molecules, 2022, doi:10.3390/molecules27134301_

Round 1

Reviewer 1 Report

The manuscript aims to proposes a virtual screening-based drug discovery to assess the efficacy of pharmacologically active compounds in green synthesized ZnO NPs loaded nanostructured oil carriers (NLC) synthesized using a mixture of aqueous seed extracts. However there are certain points that needs further clarification:

1. It is not clear why the authors did not perform computational analysis of the ZnO loaded NLCs together with the compounds. How are these compounds present in this system? Isn't there any interaction between the compounds and the ZnO and NLCs? This also means that once they are docking against the viral proteins, they should also consider this point. Because at the end, the whole system (the compounds together with the ZnO NLCs) will show antiviral activity based on their final 3D structure and possible non-covalent interactions. 

2. It is better to include the structure of ZnO, NLCs and the compounds alone (maybe as a supplementary), to better guide the reader throughout the manuscript. 

3. The resolution of Figure1D, and Figure 2 are very low. They should be replaced.

Author Response

The authors appreciate the time and effort that the editorial team and the reviewers dedicated in providing a constructive feedback on the manuscript. We are grateful for the insightful comments and valuable improvements to our paper. We have tried our best to incorporate most of the suggestions made by the reviewers. The changes have been incorporated within the manuscript. Please find below, a point-by-point response to the reviewer’s comments

Response to the comments of Reviewer 1

  1. It is not clear why the authors did not perform computational analysis of the ZnO loaded NLCs together with the compounds. How are these compounds present in this system? Isn't there any interaction between the compounds and the ZnO and NLCs? This also means that once they are docking against the viral proteins, they should also consider this point. Because at the end, the whole system (the compounds together with the ZnO NLCs) will show antiviral activity based on their final 3D structure and possible non-covalent interactions.

Response:

Computational analysis of the ZnO loaded NLCs together with the compounds is not practically feasible as there were 26 constituent compounds along with ZnO in the entire nanostructure. Therefore, similar to most of the previously published studies, individual compounds were docked to evaluate their binding efficacy with the protein. The authors do agree that there are definitely  molecular interactions between the compounds and the ZnO , NLCs as a whole. As aptly suggested, it is important to consider how the whole system  (the compounds together with the ZnO NLCs) will exhibit the antiviral activity. However, only further in vitro and in vivo studies  would facilitate  to validate the effectiveness of  the nanostructure(ZnO NLCs)  as a whole system. The key findings of the our  study  offers a prospective anti-viral drug delivery system against Covid-19.

Mohamed, J.M.M.; Alqahtani, A.; Kumar, T.V.A.; Fatease, A.A.; Alqahtani, T.; Krishnaraju, V.; Ahmad, F.; Menaa, F.; Alamri, A.; Muthumani, R.; et al. Superfast Synthesis of Stabilized Silver Nanoparticles Using Aqueous Allium sativum (Garlic) Extract and Isoniazid Hydrazide Conjugates: Molecular Docking and In-Vitro Characterizations. Molecules 2022, 27, 110. https://doi.org/10.3390/ molecules27010110

  1. It is better to include the structure of ZnO, NLCs and the compounds alone (maybe as a supplementary), to better guide the reader throughout the manuscript.

Response:

As suggested the structures of ZnO and the constituent compounds have been added as supplementary material. However, the 2D or 3D structure of the NLCs as a whole is not available. The characterization of the NLCs includes the assessment of the morphology which is based on the TEM images obtained that have been included in the results.

  1. The resolution of Figure1D, and Figure 2 are very low. They should be replaced.

Response:

Yes, the authors do agree that the resolution of Figure 1D and Figure 2 is low even though the figures are original images obtained from the devices used. Since the individual figures have been compressed into one single figure, the resolution is reduced. This was done considering the stipulated number of figures allowed as per the Author’s guidelines of the journal. However, individual images have been added for reference as supplementary material.

Reviewer 2 Report

Reviewer’s comment

 In the study titled ‘In silico studies on Zinc oxide based nanostructured oil carriers with seed extracts of Nigella sativa and Pimpinella anisum as Potential Inhibitors of 3CL Protease of SARS-CoV-2’, the authors Hendi et al., synthesized and characterized Zinc oxide nanoparticle (ZnO-NP) and done docking studies with main protease 3C of SARS-CoV-2.

ZnO-NP-loaded nanostructured oil carriers (NLC) synthesized using a mixture of aqueous seed extracts of Nigella sativa and Pimpinella anisum with olive and black seed essential oils as a potential inhibitor of 3CL protease (3CL pro). The average particle size of synthesized ZnONPs was 149.7 nm, and the polydispersity was 0.294. The authors claim potential use as anti-SARS-CoV-2 based on the docking studies of 28 phytochemicals and ZnO from nanoparticles with 3CL of SARS-CoV-2. The docking scores for best phytochemicals were -7.9 to -9.9 kcal/mol. Selected molecules (Oleuropein, δ Hederin, Dithymoquinone, and Zinc Oxide) were further characterized by Quantitative structure-activity relationship (QSAR) and molecular dynamics simulation studies.

The concepts and study look interesting, but the activity studies of ZnO-NP in a biological system are lacking to validate the prediction. However, the results go with the title as it is in-silico. There are few previous reports of extracts from the above to plant species, or the phytochemicals in them have antiviral activity. Reports of small-scale clinical trials are also available. The authors have not paid much attention to the writing part. Following are a few suggestions that might improve the value of the manuscript.

  1. The writing needs to be improved. Formatting to be done in many places, including spaces, uniformity, etc  
  2. Have the authors done docking studies with other proteases such as  PL-pro of SARS-CoV-2 before selecting 3CL as the drug target? This will tell us how specific these compounds are toward the 3CL-pro.
  3. The identified compounds are predicted to bind at three cavities of in 3CL pro. Does any of these cavities located near the active site? Or are the authors expecting allosteric inhibition? Do any compounds for which the anti- 3CL-pro activity was established bind to these cavities?
  4. The figure legends to be with details, instead of writing information on the figure.
  5. Line 342: The sentence looks incomplete. ‘ll the 27 bioactive com……
  6. Line 523-527: Consider rephrasing the sentence. Too long sentence.
  7. In docking studies, the properties of the active compounds may be compared with a compound that was reported to be tested with recombinant 3CL assays.
  8. The optimized compounds may be active when tested in pure form. But the concentration in the extract or nanoparticle may not be sufficient to inhibit SARS-CoV-2. If the ZnO nanoparticle is dosed at higher concentrations, there can be an issue of toxicity.  
  9. The conclusion looks too general. Need to add what kind of future studies are required.
  10. Line 493: 4.5. Molecular dynamics simulation --To be corrected as 4.6

Author Response

Dear Editor,

Thank you for giving us the opportunity to submit a revised draft of the manuscript titled  “In silico studies on Zinc oxide based nanostructured oil carriers with seed extracts of Nigella sativa and Pimpinella anisum as Potential Inhibitors of 3CL Protease of SARS-CoV-2’’ for potential publication in your journal. We appreciate the time and effort that the editorial team and the reviewers dedicated in providing a constructive feedback on the manuscript. We are grateful for the insightful comments and valuable improvements to our paper. We have tried our best to incorporate most of the suggestions made by the reviewers. The changes have been incorporated within the manuscript. Please find below, a point-by-point response to the reviewer’s comments

Response  to Reviewer 2

  1. The writing needs to be improved. Formatting to be done in many places, including spaces, uniformity, etc  

Response: The MS has been thoroughly checked for grammatical errors, and formatting.

  1. Have the authors done docking studies with other proteases such as  PL-pro of SARS-CoV-2 before selecting 3CL as the drug target? This will tell us how specific these compounds are toward the 3CL-pro.

Response: Well,the authors did not do any previous docking studies with other proteases such as  PL-pro of SARS-CoV-2. The particular viral protein target(3CL-pro) was selected based on the previous molecular docking studies which have used similar compounds from Nigella sativa and Pimpenella anisum seeds and Zn for the same protease. However, docking studies with other proteases such as  PL-pro  of SARS-CoV-2  could be a part of the prospective future studies by the authors.

Su, Hx., Yao, S., Zhao, Wf. et al. Anti-SARS-CoV-2 activities in vitro of Shuanghuanglian preparations and bioactive ingredients. Acta Pharmacol Sin 41, 1167–1177 (2020). https://doi.org/10.1038/s41401-020-0483-6

  1. The identified compounds are predicted to bind at three cavities of in 3CL pro. Does any of these cavities located near the active site? Or are the authors expecting allosteric inhibition? Do any compounds for which the anti- 3CL-pro activity was established bind to these cavities?

Response: These binding cavities are located near the substrate-binding active site. This can be best explained based on the substrate-binding pocket represented by an intermolecular surface in a study by Su et al.,2020.I t was reported that the protease had a catalytic Cys145-His41 dyad and an extended binding site, features shared by SARS-CoV 3CLpro and MERS-CoV 3CLpro.

Examination of the active site of the complex revealed that three phenolic hydroxyl groups of baicalein made multiple hydrogen bonds with the main chains of Leu141/Gly143 as well as the side chains of Ser144/His163. The only carbonyl group established a hydrogen bond with the main chain of Glu166, hydrophobic interactions were found to be with multiple residues Gln189/Arg188/Met49/Cys44/His41.Similar,  ligand-protein interactions with above mentioned amino acids were observed in our study as well. 

Thus, the inhibition is not allosteric but competitive inhibition.

(Su et al.,2020)

Su, Hx., Yao, S., Zhao, Wf. et al. Anti-SARS-CoV-2 activities in vitro of Shuanghuanglian preparations and bioactive ingredients. Acta Pharmacol Sin 41, 1167–1177 (2020). https://doi.org/10.1038/s41401-020-0483-6

  1. The figure legends to be with details, instead of writing information on the figure.

Response: The figure legends have been revised as suggested.

  1. Line 342: The sentence looks incomplete. ‘ll the 27 bioactive com……

Response: Corrected

  1. Line 523-527: Consider rephrasing the sentence. Too long sentence.

Response: The sentence has been rephrased.

  1. In docking studies, the properties of the active compounds may be compared with a compound that was reported to be tested with recombinant 3CL assays.

Response: The highlighted top scored compounds in our study based on the docking score against the 3CL-pro were in the order of Delta-Hederin >Zinc oxide>Oleuropein>Dithymoquinone. The range of the docking score was -9.9kCal/mol to -7. 9kCal/mol was close to certain compounds reported in a recent study. Active compounds with ΔG less than -8.0 Kcal/mol were selected, namely Quinine, Quinidine, Cinchonine, and Lovastatin in a previous study by Haniyya et al.(2022). Thus, the docking score of the optimized compounds in our study were in line with those standard compounds tested with recombinant 3CL assays.

Citation: Haniyya et al 2022 IOP Conf. Ser.: Earth Environ. Sci. 976 012051. https://doi.org/10.1088/1755-1315/976/1/012051

  1. The optimized compounds may be active when tested in pure form. But the concentration in the extract or nanoparticle may not be sufficient to inhibit SARS-CoV-2. If the ZnO nanoparticle is dosed at higher concentrations, there can be an issue of toxicity. 

 Response: Well, the authors do agree that the present study only introduces a prospective use of the nanoparticles as a potential anti-viral drug and provides ideal leads towards exploring ex vivo, in vitro and in vivo studies. Only these further studies can validate the  actual efficacy of  ZnO nanoparticles at a particular concentration after standardizing the optimal concentration in experimental trials.

  1. The conclusion looks too general. Need to add what kind of future studies are required.

Response: Conclusion has been rewritten.

  1. Line 493: 4.5. Molecular dynamics simulation --To be corrected as 4.6.

Response: Corrected.

Round 2

Reviewer 2 Report

Title: In silico studies on Zinc oxide based oil carriers 2 with seed extracts of Nigella sativa and Pimpinella anisum as 3 Potential Inhibitors of 3CL Protease of SARS-CoV-2

Authors: Hendi et al.,

Reviewers comment:

1.      The authors’ response is satisfactory but in the revised manuscript there is not much improvement in formatting especially spacing.  The following are just examples.  Line 64, 67, 70, 99, 104, 132, 134, 173, 174, 202-205, 213-215, 220, 243, 244, 260, 261, 269, 270, 300-302, 420.    

2.      The authors may consider including the summary of the author's response to comments 3 and 7 for the original version may be included in the discussion.

3.      Line 35: Expand COVID-19 and the expansion in line 43 to be removed.

4.      Line 41-42.  Globally, there …… reported to WHO. –  As of which date? Also, include references. 

Author Response

Response to the Reviewers comments:

  1. The authors’ response is satisfactory but in the revised manuscript there is not much improvement in formatting especially spacing.  The following are just examples.  Line 64, 67, 70, 99, 104, 132, 134, 173, 174, 202-205, 213-215, 220, 243, 244, 260, 261, 269, 270, 300-302, 420.  

Response: Formatting has been improved as suggested.  

  1. The authors may consider including the summary of the author's response to comments 3 and 7 for the original version may be included in the discussion.

Response: The response to comments 3 and 7 has been included with references.

  1. Line 35: Expand COVID-19 and the expansion in line 43 to be removed.

 Response: Corrected.

  1. Line 41-42.  Globally, there …… reported to WHO. –  As of which date? Also, include references. 

Response: Corrected as suggested.